# Novel Aspects of cAMP-Response Element Modulator (CREM) Role in Spermatogenesis and Male Fertility

**DOI:** 10.3390/ijms241612558

**Published:** 2023-08-08

**Authors:** Diego Eduardo Sánchez-Jasso, Sergio Federico López-Guzmán, Rosa Maria Bermúdez-Cruz, Norma Oviedo

**Affiliations:** 1Departamento de Genética y Biología Molecular, Centro de Investigación y de Estudios Avanzados del Instituto Politécnico Nacional (CINVESTAV-IPN), Mexico City 07360, Mexico; diego.jasso@cinvestav.mx (D.E.S.-J.); sergio.lopez@cinvestav.mx (S.F.L.-G.); roberm@cinvestav.mx (R.M.B.-C.); 2Unidad de Investigación Médica en Immunología e Infectología, Centro Médico Nacional La Raza, Instituto Mexicano del Seguro Social (IMSS), Mexico City 02990, Mexico

**Keywords:** CREM, spermatogenesis, male fertility, gene regulation, CREM isoforms

## Abstract

Spermatogenesis is a very complex process with an intricate transcriptional regulation. The transition from the diploid to the haploid state requires the involvement of specialized genes in meiosis, among other specific functions for the formation of the spermatozoon. The transcription factor cAMP-response element modulator (CREM) is a key modulator that triggers the differentiation of the germ cell into the spermatozoon through the modification of gene expression. CREM has multiple repressor and activator isoforms whose expression is tissue-cell-type specific and tightly regulated by various factors at the transcriptional, post-transcriptional and post-translational level. The activator isoform CREMτ controls the expression of several relevant genes in post-meiotic stages of spermatogenesis. In addition, exposure to xenobiotics negatively affects *CREMτ* expression, which is linked to male infertility. On the other hand, antioxidants could have a positive effect on *CREMτ* expression and improve sperm parameters in idiopathically infertile men. Therefore, *CREM* expression could be used as a biomarker to detect and even counteract male infertility. This review examines the importance of CREM as a transcription factor for sperm production and its relevance in male fertility, infertility and the response to environmental xenobiotics that may affect *CREMτ* expression and the downstream regulation that alters male fertility. Also, some health disorders in which *CREM* expression is altered are discussed.

## 1. Introduction

Spermatogenesis is regulated by signals from the hypothalamic–pituitary–gonadal axis. Upon the onset of puberty, this axis triggers the release of luteinizing hormone (LH) and follicle-stimulating hormone (FSH) [1]. LH reaches the Leydig cells in the testes, stimulating testosterone production. Testosterone then binds to androgen receptors on Sertoli cells, Leydig cells and peritubular myoid cells, and initiates signals that stimulate the spermatogonia differentiation pathway, introducing mature sperm [2]. Spermatogenesis is highly sensitive to testosterone released by Leydig cells and retinoic acid (RA) synthesized by Sertoli cells. After that, germ cells are able to synthesize their own RA [3]. During this process, the expression of exclusive germ cell genes involved in several processes like meiosis, DNA condensation and spermiogenesis are regulated by testis specific transcription factors. Also, some transcripts even undergo translation during maturation in epididymis and sperm capacitation [4,5].

The discovery of the cAMP-response element modulator (CREM) as the major transcription factor for murine spermatogenesis is of great importance. It is able to produce more than 30 isoforms—some of them displaying activating functions and an enhanced expression in the testis [6]. The various isoforms of *Crem* are produced through molecular events such as transcription from alternative promoters, alternative splicing, alternative polyadenylation and the involvement of destabilizing elements at the 3′ end of RNA [7]. Upon the onset of puberty, the pituitary hormone FSH initiates the expression of *Cremτ* (activator isoform) [8]. This isoform consists of two Q-rich regions, a Kid domain (kinase-inducible), a basic region, an NLS signal for nuclear import and a leucine zipper at its carboxyl end. CREM has the ability to homodimerize (Figure 1; Appendix A) and heterodimerize with other transcription factors via its leucine zipper domain. Crem recognizes palindromic sequences known as CRE sites with a consensus sequence of TGACGTCA and also an only 4-bp-long sequence known as half-CRE sites. These sequences can also be recognized by cAMP-response-element-binding protein (CREB) [9,10].

In somatic cells, the activation of CREB and CREM proteins involves phosphorylation and their interaction with coactivator proteins such as CREB-binding protein (CBP) and p300 [11]. However, unlike CREB, CREM*τ* does not require phosphorylation for activation in germ cells. Instead, its activation occurs through another coactivator protein called Activator of CREM in Testis (ACT) to modulate its nuclear transcriptional activity [12]. ACT contains a LIM domain with two zinc fingers likely to facilitate protein–protein binding [13]. Interestingly, a kinesin protein (KIF17b) regulates ACT also present in germ cells. KIF17b is responsible for clearing ACT from the cell nucleus, thereby preventing the activating function of CREM*τ* [14].

Despite silencing *Act*, CREM-dependent genes involved in sperm fertility maintenance continue to be expressed, and instead, the affected gene group is primarily associated with a decrease in sperm count and flagellum motility [15].

An intriguing mechanism involves the coordinated process of transcription, KIF17b facilitated cytoplasmic transport and the translation of Crem-dependent gene mRNAs. This mechanism is mediated by a set of RNA-binding proteins that promote the protection and span of the mRNA half-life of CREM-dependent genes in the cytoplasm [16]. It is important to note that most of the knowledge about CREM involvement in spermatogenesis was generated from 1992 to 2006 and several reviews released in this period describe essential aspects of CREM. However, little information about the role of CREM in spermatogenesis has been reported beyond 2006 and no work has reviewed *state of the art* investigations in recent years. In this review, a compilation of recent contributions is discussed and highlighted to explore different molecular aspects of CREM regulation in reproduction, the modulation of its activity and the implications of its deregulation on spermatogenesis and fertility.

## 2. CREMτ Expression

It is widely accepted that the up regulation of *Crem* expression during spermatogenesis occurs in response to LH and FSH signaling [8]. These hormones bind to specific receptors on Leydig and Sertoli cells, leading to an increase in cAMP levels and subsequent changes in gene expression. FSH and forskolin induce an elevated expression of *Icer* (Inducible cAMP Early Repressor) in rat Sertoli cells. When the ICER protein level increases, it undergoes autoregulation, resulting in a decrease in its expression [17]. Repressor isoforms are initially expressed at low levels in early spermatogonia and spermatocytes. The expression of the activating isoform *Cremτ* in spermatocytes triggers the expression of meiotic genes crucial for gamete differentiation into spermatozoa. The human *CREM* gene is located on chromosome 10 (86,108 bp) and consists of eight exons, of which seven exons correspond to the coding sequence of the mRNA (2643 bp) encoding the isoform CREMτ (332 amino acids). In contrast, the mouse *Crem* gene is located on mouse chromosome 18 (74,504 bp) and contains nine exons, producing a 2362 bp mRNA for the 357aa CREMτ isoform. The mouse *Crem* gene shares an 86.03% identity with the human gene and encodes a CREMτ protein that shares an 86.63% identity with its human counterpart.

To date, 54 transcript isoforms of the human *CREM* gene, independently of a genome build, have been identified that produce mRNAs ranging from 925 bp to 3500 bp. However, only 17 isoforms and proteins are recognized in specific genomic databases. In contrast, the murine *Crem* gene has 42 transcript isoforms in the annotated genome GRCm39 C57BL/6J with 18 confirmed isoforms ranging from 529 bp to 2931 pb (Figure 2).

The similarity between the *Creb* and *Crem* genes lies in several aspects. Firstly, their first exon does not contain a coding sequence (CDS), and the start translation codon is located in the second exon. In addition, both proteins possess two glutamine-rich activation domains and a P-box site for phosphorylation by PKA, encoded by two exons. The carboxyl terminus of both proteins contains a basic region containing the leucine zipper domain (bZIP) (Figure 2), which allows for DNA binding and homodimerization as well as heterodimerization [7].

The phosphorylation of serine-117 of CREMτ by testis-specific kinase activity in vitro, dependent on changes in cAMP and not the Ca_2_-calmodulin pathway, has been suggested as a potential regulatory mechanism for the protein [18]. However, it has been demonstrated that the activation of CREMτ during spermatogenesis does not depend on phosphorylation but rather on the presence of Act, which serves as a biological marker for male fertility [12].

Before the expression of the *Cremτ* isoform in testes, various dominant-negative proteins of CREB, such as CREBγ and CREBα-γ isoforms, are expressed. These isoforms lack the ability to bind to DNA or transactivate genes, but they modulate other CREB isoforms. Low levels of the CREB isoforms δ and α (transactivators) are expressed in primary spermatocytes [19]. The *Crem* gene encodes different isoforms, including both repressor and activator isoforms. Among the repressors are CREM α, β, γ, CREM-S and ICER, while the isoforms CREMτ 1 and 2 are identified as activating transcription factors essential for spermatogenesis [18]. The CREMΔC-G isoform, which lacks the two glutamine-rich activation domains and the P-box, acts as a repressor and competitor for CRE sites in elongated spermatids (Figure 2) [20].

Furthermore, the presence of alternative promoters P3 and P4 allows for the expression of different *Cremθ1* and *2* isoforms, along with alternative splicing and translation initiation, which contribute to the generation of various isoforms of the gene, including *Cremα*, *β*, *γ* and *Crem-S* [7,21,22,23]. In low-bred sheep, alterations in *Crem* alternative RNA splicing were observed, resulting in phenotypic changes associated with infertility. These changes are related to alterations in *Crem* isoforms that control the expression of genes closely involved in spermatogenesis [24]. The alternative polyadenylation of *Crem* mRNA is another element contributing to the wide variety of isoforms and their stability. The presence of a testis-specific paralog of the stimulatory cleavage factor (CSTF), called τCST-64, facilitates the interaction of the polyadenylation-specific factor (CPSF) with the downstream sequence element (DSE) of *Cremτ2* mRNA in the meiotic stage (Figure 2). This interaction promotes cleavage at specific polyadenylation sites, such as site 3 out of 11 polyadenylation sites. The absence of τCST-64 impairs polyadenylation at site 3 and enhances polyadenylation at site 8, leading to the deregulation of sperm morphology and motility, ultimately resulting in infertility in mice [25].

Both human and murine genes share their promoter region with divergent genes (Cullin 2 (*CUL2*) and *Gm6225*), and they exhibit a similar expression profile in different tissues, suggesting the presence of a bidirectional promoter within the *Crem* promoter region and its first intron. Additionally, there are two enhancers downstream of the *CREM* gene. One enhancer is associated with NANOG and is marked by modified histone H3K27ac, while the other is labeled with modified histone H3K4me1 [26,27].

Single-cell RNA sequencing data confirm that *Cremτ* expression in mouse spermatocytes begins depending on the pachytene spermatocytes and reaches its peak expression in stage-VII-to-VIII round spermatids but is absent in elongated spermatids (Figure 3) [28]. A similar expression profile of human *CREMτ* has been observed in normozoospermic males, while oligozoospermic males only maintain the expression of repressor isoforms [29]. However, it should be noted that the expression and localization of CREMτ orthologs might differ in other species; in porcine spermatids and spermatozoa, CREMτ is still detected around the cell nucleus and in the connecting piece, respectively [30].

The expression of CREMτ and its coactivator ACT has been confirmed in human testicular biopsies from patients with obstructive azoospermia (OA), as has its correlation with the expression of transition proteins 1 and 2 (TNP1, 2) and protamines 1 and 2 (PRM1, 2), which are required for the packaging of DNA [31]. Instead, in patients with Sertolli-Cell-Only Syndrome (SCOS), all of these genes are down-regulated, but even in the absence of *CREM*, a low expression of *TNP2* is still seen. Infertile patients diagnosed with arrested round spermatid maturation (SMA) have a low expression of *CREM*τ and *ACT*, which negatively correlates with the expression of these genes [31]. These findings are consistent with previous reports on mice and indicate that a reduced expression of *Crem* and its cofactor *Act* is associated with post-meiotic arrest and male infertility. The detection of human CREMτ in different tissues shows that its main expression occurs in the testis. It is also expressed in other tissues such as the adrenal gland, placenta, appendix and bladder, reaching levels greater than or approximately 100 transcripts per million [32] in contrast to mouse tissues, which, with the exception of the testis, have a low expression.

In patients with obstructive azoospermia, the expression of CREMτ and ACT correlates with the expression of genes involved in chromatin compaction (TPN, PRM1 and PRM2). Infertile patients with post-meiotic arrest and the persistence of round spermatids exhibit a low expression of *CREM*τ and *ACT*, which negatively correlates with the expression of these genes. These findings are consistent with previous reports on mice and indicate that a reduced expression of *Crem* and its cofactor *Act* is associated with post-meiotic arrest and male infertility [31]. When examining the expression of *CREM* in different cancers, over-expression has been observed in skin and immune cancers, whereas down-regulation is observed in other types of cancers. However, the detection of CREM protein, which has a well-conserved DNA binding domain among the 28 isoforms recognized, is ambiguous, as the detection may correspond to both repressor and activator isoforms, affecting the interpretation of CREM expression in different tissues [32].

## 3. CREM Regulation

CREM is widely recognized as a crucial regulator of spermatogenesis; therefore, the comprehension of its modulation at all levels is relevant. A well-established aspect is the direct regulation of CREM activity by ACT-KIFL7b, while the involvement of other ACT-like factors is still largely unclear (see [16] for the latest review). This section focuses on analyzing recent reports on new CREM modulators at transcriptional, post-transcriptional and post-translational levels and their mechanisms, which are summarized in Figure 4.

### 3.1. Transcriptional Regulation

Follicle-stimulating hormone (FSH) is a gonadotropin that specifically binds to its receptor on Sertoli cells in the testicles. It has been suggested that FSH plays a role in the switching and maintenance of *Crem* mRNAs, since a reduction in *Crem* transcripts in hypophysectomized rats and an increase following direct FSH injection has been observed [8]. However, the specific factors responsible for the association between FSH and Crem regulation remain unclear. On the contrary, bona fide regulation of Crem expression by FSH has been strongly challenged as mice lacking FSH receptor are still fertile with normal CREM expression [33,34]. Furthermore, testicular cells from rats and macaques under gonadotropin releasing hormone (GnRH) deficiency also show normal CREM expression [10]. Recently, proper FSH signaling has been shown to be critical for the maintenance of the spermatogonial stem cell pool and differentiation and initiation of meiosis [35,36]. This raises questions on whether a gated FSH-*Crem* expression mechanism exists or is simply a consequence of changes in testicular cell population or some other unrelated effects that have not been previously studied. Therefore, the detection of testicular cell populations must be performed during a histological analysis of hypophysectomized mice along with detecting *Crem* expression or protein immunodetection and other specific markers for each cell type. These analyses are fundamental to confirm or reject a direct regulatory mechanism of FSH on *Crem* expression.

Death-Associated Protein-like 1 (DAPL1) is a key regulator of testosterone production by Leydig cells. Mice with *Dapl1* deficiency (*Dapl1^−/−^*) exhibit an up-regulation of *Crem*, *Creb1*, *Act*, CREB-regulated transcription coactivator 1 (*Crtc1*), protein kinase C alpha (*Prkacα*) and mitogen-activated protein kinase (*Mapk3/1*) genes in the whole testis. Accordingly, an over-expression of *Dapl1* in an in vitro model of I-10 (mouse Leydig testicular tumor cells) leads to a down-regulation of the same genes [37]. Since the cAMP signaling pathway is controlled by CREB/CREM-PKA, DAPL1 is proposed as a negative regulator of cAMP signaling. However, further experiments are needed to confirm whether *Crem* is directly regulated by DAPL1.

In male fertility diseases and impaired spermatogenesis, several genes are differentially methylated at the DNA level [38]. Surprisingly, patients with abnormal protamination and oligozoospermia showed significantly higher *CREM* promoter methylation than normal controls [39,40]. Although the analysis of *Crem* expression is lacking in these studies, experiments using mouse spermatocyte GC-2spd(ts) cells have shown that the hypermethylation of the *Crem* gene is significantly associated with decreased expression levels of its transcript [41]. Furthermore, a low expression of CREM as well as a low recruitment into its target genes, including PRM1 and PRM2, have been reported in patients with the maturation arrest of round spermatids (SMA) if compared to OA patients [31]. Therefore, it is conceivable that the hyphermethylation of the *CREM* promoter leads to a reduction in *CREM* expression, leading to the reduced expression of protamine genes that ultimately causes the abnormal protamination observed in patients.

Information regarding the direct transcriptional regulation of *Crem* through spermatogenesis is poor. *Crem* contains multiple promoters that generate distinct isoforms (Figure 3); its expression is differential and tissue-cell specific; and it is essential for male fertility. Therefore, it is important to note that our understanding is currently limited to the general cause and effect approach to *Crem* expression, and we lack information on precise mechanisms or models by which *Crem* expression is directly regulated during spermatogenesis.

### 3.2. Post-Transcriptional Regulation

The spatial and temporal control of CREM activator and repressor isoforms is regulated during spermatogenesis to ensure accurate gene expression. It has been reported that FSH appears to regulate the expression of *Crem* isoforms and enhances *Cremτ* mRNA stabilization through the use of alternative polyadenylation sites [6]. However, the validity or specific mechanisms by which FSH could drive a *Crem* expression switch are yet to be identified.

τCSTF-64 is an important RNA-binding protein in the cleavage and polyadenylation of mRNAs during meiosis and later stages of spermatogenesis. Testes from *τCstF-64*^−/−^ mice show a reduced expression of the *Cremτ2* isoform due to the preferential use of a distal cleavage/polyadenylation site in exon 3 (P3) instead of the proximal site used in Wt mice, resulting in a reduced expression of 15 CREM target genes. Therefore, it has been suggested that τCSTF-64 mediates exon 4 (Q1 domain) exclusion and contributes to the infertility phenotype of *τCstF-64*^−/−^ mice by increasing exon 4-containing isoforms [25]. Nevertheless, these *τCstF-64*^−/−^ mice exhibit reduced numbers of round spermatids, which makes it possible that alteration on *Cremτ* isoforms is due to lacking cells expressing the corresponding isoforms. On the other hand, what is deleted in azoospermia-associated protein 1 (*Dazap1*), an hnRNP abundantly expressed in the testes, produces the opposite effect of τCstF-64. Testes from *Dazap1*^−/−^ mutant mice display an overall decrease in *Crem* mRNA levels, but also exhibit a significant decrease in the percentage splicing inclusion (PSI) of exon 4, leading to a notable increase in exon 4-free isoforms. Furthermore, the ectopic expression of Dazap1-Wt results in a significant increase in *Crem* exon 4 inclusion but mutant mice failed to restore this isoform. The precise mechanism by which Dazap1 mediates exon inclusion/exclusion is not fully understood. However, Dazap1 can facilitate the inclusion of exon 4 of *Crem* isoforms through directly binding to intron 3 [42], although its effect on expression levels of CREM target genes has not been studied. However, adult *Dazap1*^−/−^ mice do not develop any post-metiotic cells, leading to similar concerns as mentioned for *τCstF-64*^−/−^ mice, although ectopic *Dazap1* expression supports its role in *Crem* exon 4 inclusion.

Analyzing *Crem* isoform expression and its impact on the expression of target genes is relevant as it would help to identify the roles of specific domains under natural conditions. ACT-mediated activation can only occur in CREM-activating isoforms that contain both a P-box domain and at least one Q domain [12]. A reduction in the activating *Cremτ2* isoform, lacking the Q1 domain, results in a reduction in mRNA levels of 15 CREM target genes [25]. This suggests that the Q1 domain may play a specific role in mediating the activation of these genes, possibly through interactions with different cofactors. Further experimental evaluation is required to determine whether CREMτ Q1 and Q2 domains are directly involved in the differential expression of their target genes.

### 3.3. Regulation of Protein Activity

CREM protein activity is tightly regulated through different mechanisms but recent reports show negative regulation as the major one, mainly by sequestering cofactors needed by CREM to activate gene expression.

*Jmjd1a* encodes H3K9me1/H3K9me2 lysine demethylase, which is mainly expressed in pachytene and second spermatocytes [43]. Knockout of *Jmjd1a* in male mice (*Jmjd1a^−/−^*) results in severe oligozoospermia and a complete sterile phenotype, similar to *Crem*^−/−^ mice. Although a JMJD1a deficiency does not affect *Crem* mRNA levels, it reduces the mRNA expression of its coactivator ACT. Consequently, known Crem target genes (*Tnp1*, *Tnp2*, *Prm1*, *Prm2*, outer denser fiber of sperm tails 1 (*Odf1*) and capping actin protein of the muscle Z-line (*Gsg3*)) exhibit a significantly reduced expression in *Jmjd1a*^−/−^ mice due to an increased methylation of their promoter regions, a down-regulation of *Act* mRNA and the subsequent reduction in CREM occupancy. Thus, JMJD1a indirectly modulates the CREM protein activity, enhances the expression of ACT and maintains low levels of methylation on its target gene promoters, allowing for the correct positioning and functioning of CREM [43]. Similarly, reduced mRNA levels of such CREM coactivators as four and a half LIM domains 5 (*Fhl5*) and *Kif17* are observed in τ*CstF-64*^−/−^ mice, resulting in the down-regulation of some direct CREM targets (*Prm1*, *Tssk1*, *Tssk2*, *Tnp1*, *Tnp2* and heat shock protein family A member 1 (*Hspa1l*)) [25]. Therefore, τCSTF-64 is believed to be an indirect regulator of CREM activity, but the underlying mechanism remains unclear.

Protein serine/threonine kinase 4 (TSSK4) is a testis-specific kinase that, in its active (phosphorylated) form, can phosphorylate serine 133 of CREB and has a role in the cAMP response pathway during spermatogenesis [44]. In fact, the expression pattern of *Tssk4* in testis overlaps almost perfectly with that of *Crem*. Moreover, a direct interaction between TSSK4 and CREM has been demonstrated in in vitro assays. Furthermore, TSSK4 can efficiently phosphorylate CREM serine 117 and enhance its activity in vitro, thus raising the possibility of TSSK4 enhancing CREM activity in post-meiotic states through a phosphorylation-dependent pathway [45]. However, in order to demonstrate TSSK4′s role as a CREM regulator, an assessment of the interaction and phosphorylation of CREM using TSSK4 in vivo is required, along with its effect on CREM target gene expression.

CARM1 (PRMT4) is a protein arginine methyltransferase that is vital for postnatal survival. It exhibits a high expression in the testis, particularly during late stages of spermatid formation, overlapping CREM expression. Germline conditional knockout has shown that male *Carm1*-cKO mice are infertile due to a significant reduction in the number of elongated spermatids. CBP/p300 is a natural substrate of CARM1. Indeed, CARM1 has been shown to methylate p300 protein, preventing its interaction with the first LIM domain of ACT and reducing CREMτ transactivation [46]. Bao et al. hypothesizes that ACT recruits p300 coactivators to activate germline-specific gene expression in early stages of round spermatids. By inhibiting the interaction between p300/ACT/CREMτ, the expression of genes bound by CREMτ/ACT in elongated spermatids is reduced, ensuring the appropriate metabolic program for normal spermatid development.

Sperm-associated antigen 8 (*Spag*8) is specifically expressed in the testis and is localized in the nucleus and cytoplasm of spermatocytes and round spermatids, while it is found exclusively in the cytoplasm of elongated spermatids [47]. SPAG8 can directly interact with ACT in vitro through two domains in SPAG8 with the second LIM domain of ACT. The SPAG8-ACT interaction can support the binding of ACT to CREMτ, but without forming a SPAG8-ACT-CREMτ complex, thereby increasing the transcriptional activity of CREMτ. Furthermore, the SPAG8-ACT interaction is independent of CREMτ S117 phosphorylation, as previously reported [47]. It is clear that SPAG8 plays a role in ACT-CREMτ-mediated transcriptional regulation, but the underlying mechanism remains unknown. It has been suggested that regulation may involve the microtubule-dependent nuclear transport of SPAG8, facilitating ACT translocation into the nucleus in round spermatids. This possibility has been previously explored in the nuclear transport of ACT by KIF17b, although this transport is independent of its motor domain or microtubule transport, but it is dependent on KIF17b phosphorylation, possibly with protein kinase A [14,48]. Therefore, it is possible that the regulation of CREMτ by SPAG8 follows a similar mechanism.

Tudor domain protein 5 (TDRD5) plays a critical role in the proper assembly of cytoplasmic ribonucleoprotein granules known as intermitochondrial cements (IMCs) and chromatoid bodies (CBs), which are involved in RNA processing during spermatogenesis [49]. Male *Tdrd5*^−/−^ mice are sterile due to arrest in the post-meoitic round spermatids with a complete absence of elongated spermatids and spermatozoa (resembling the CREM phenotype). Interestingly, *Tdrd5*-deficient round spermatids exhibit comparable levels of *Crem* expression to wild-type cells, but the expression of *Act* mRNA and CREM targets *Prm1*, *Prm2*, *Tnp1* and the major sperm tail component *Rt7* is significantly reduced [49]. Thus, the failure of spermiogenesis in *Tdrd5*-deficient cells is likely a consequence of a reduced expression in CREM target genes in round spermatids, potentially due to a reduction in *Act* mRNA, as well as a defective function of IMCs and CB in post-meiotic cells.

Germ cell nuclear factor (GCNF) is a highly expressed transcription factor in spermatids. It physically interacts with CREMτ and competes for binding to a CRE/nuclear receptor binding site (CRE/NR), thereby repressing its transcriptional activity. Although, a deletion assay suggests that repression does not entirely depend on the GCNF-CREMτ interaction but rather also on DNA binding. GCNF enhances the deacetylation of histone H3 or at least prevents acetylation by CREMτ without affecting its established promoter binding. Moreover, the expression of five reporter plasmids containing CREM target promoters is blocked by the expression and co-expression of GCNF, indicating a clear repression activity. The functionality of the CRE/NR site in vivo is evident, as it can drive the testis-specific expression of a reporter gene, particularly in post-meiotic spermatids [50]. Therefore, GCNF-CREMτ competition could be a general mechanism regulating testicular genes, but the relationship between GCNF-CREMτ and CRE/NR sites in multiple-gene expression requires further investigation.

ACT-deficient mice (*Fhl5*^−/−^) can progress through the post-meiotic phase and generate spermatozoa, albeit with a drastic reduction in the sperm count and an increase in structural abnormalities. This suggests that not all CREM target genes require ACT, but those encoding structural genes essential for mature sperm do [15]. Although it was previously reported that there is no difference in gene expression levels in the testes of Fhl5^−/−^ and wt mice, these analyses are highly questionable due to the sensitivity of microarray assays and the potential influence of genetic background on testicular expression profiles [51]. Additionally, accumulating evidence links a decreased expression or altered subcellular localization of ACT to the infertility phenotype observed in *Crem^−/−^* mice.

The regulation of Crem activity at the protein level is one of the most extensively studied aspects to date, given its role as a master regulator during spermatogenesis. Taken together, the data indicate that JMJD1a, τCSTF-64, TSSK4, CARM1, SPAG8, TDRD5 and GCNF are novel modulators of CREM activity, either directly or indirectly.

## 4. Genes Regulated by CREM

The critical role of CREMτ in spermatogenesis has been highlighted since the generation of *Cremτ*-deficient mice. These mice exhibit a decreased expression of testis-specific genes, including *Prm1*, *Prm2*, *Tnp1*, *Tnp2*, sperm-mitochondria-associated cysteine-rich protein (*Mcs*), *Rt7*, high-affinity Ca2+/calmodulin-binding protein *Calspermin* (*Camk4*) and early growth response *Egr2* (*Krox-20*) and *Egr1* (*Krox-24*), being the first reported set of genes regulated by CREM [52,53]. Over time, a growing number of genes have been proposed as potential targets of CREMτ regulation.

### 4.1. CREM Binding and Gene Expression Association

The most comprehensive analysis of differential gene expression in testes from *Crem*-knockout (KO) mice was conducted by Kosir et al. in 2012, and their study is the only one for which raw data are available. They reported a total of 4706 genes that exhibited differential expression, with 1822 genes down-regulated and 2884 genes up-regulated. However, it is important to note that a significant number of these genes are likely to be regulated indirectly, since in the absence of CREM, the expression of about 85 TFs is altered, with approximately 40 TFs directly bound to CREM. In total, they were able to identify around 1607 genes to which CREM binds directly [54,55]. But when applying a log2FC cut-off of 1, a total of 1098 deregulated genes are identified, with the majority (824 genes) showing deregulation, contrary to the findings reported by Kosir et al. [55]. Importantly, out of the 1098 genes, only 354 genes (32.24%) are bound by CREM. Notably, only down-regulated genes are primarily associated with spermatogenesis, whereas the up-regulated genes (274 genes) are involved in the response to xenobiotic stimuli and the processing of oxoacids, carboxylic acids and xenobiotics, among others (Appendix A). Of all the down-regulated genes, the gene ontology (GO)–biological processes of the genes bound by Crem are more diverse than the overall set of genes, which have a higher relative representation in GO terms related to spermatogenesis (Figure 5, Appendix A), possibly reflecting arrest in the round spermatid phenotype. Among the down-regulated genes, those involved in fertilization or spermatogenesis represent approximately 90 genes, with at least 35 gene promoters being directly bound by CREM (Table 1).

To establish an association between protein-gene binding and gene expression, it is valuable to integrate extensive expression data from techniques like microarrays or RNA sequencing in CREM-deficient and Wt models with binding data obtained through methods such as chromatin immunoprecipitation sequencing (ChIP-seq) in Wt animal models. For instance, CREMτ has been shown to bind to the phosphoglycerate kinase 2 (*Pgk2*) promoter in the testis, which starts expression at the same stage as CREMτ [56]. Accordingly, a reduction in *Pgk2* expression has been observed in *Crem*-KO mice and it suggests a potential direct regulation of *Pgk2* by CREMτ [56]. However, further experimental investigations are necessary to definitively confirm whether it is binding that indeed promotes the expression of *Pgk2*. It is crucial to conduct proof-of-concept assays to establish a causal relationship between protein binding and the regulation of gene expression.

Although the absence of CREM in vivo can lead to the deregulation of up to 4706 genes, currently available *Crem*-KO mouse models exhibit an arrest in the initial stage of round spermatids [52,53]. Therefore, we cannot solely rely on these expression data to accurately estimate the number of deregulated genes that initiate expression immediately downstream stages, nor can we ascertain whether these changes are a direct effect of a CREMτ absence. While it is possible to obtain testicular samples from patients or mice displaying the same phenotype, a CREM absence makes it impossible to identify changes in its expression and binding after first-round spermatid stages. This limitation hinders a comprehensive understanding of CREM’s control over gene expression in post-meiotic stages. Therefore, developing a model for *Cremτ* gene inactivation specifically targeting the initial post-meiotic stage would be highly beneficial.

**Table 1 ijms-24-12558-t001:** Deregulated genes involved in spermatogenesis process from CREM−KO testis. * Data from GSE29593 [55]; ** [57].

Gene Symbol	Gene ID	Bound by CREM *	Log2FC *	Gene Symbol	Gene ID	Bound by CREM *	Log2FC *
Ace	11421	Yes	−2.42	Prm3	19120	No	−4.74
Cdyl	12593	Yes	−2.72	Rxra	20181	No	−1.24
Cftr	12638	Yes	−1.33	Spam1	20690	No	−3.09
Crem	12916	Yes	−1.71	Tle3	21887	No	−1.55
Dbil5	13168	Yes	−2.91	Tssk1	22114	No	−1.56
Hspa1l	15482	Yes	−1.81	Atp1a4	27222	No	−4.83
KIf17	16559	Yes	−2.13	Prdx4	53381	No	−2.54
Prm1	19118	Yes	−5.53	Fscn3	56223	No	−4.86
Tnp1	21958	Yes	−4.86	Tssk3	58864	No	−4.65
Tnp2	21959	Yes	−5.46	Efcab1	66793	No	−3.57
Tssk2	22115	Yes	−4.15	Txndc8	67402	No	−4.72
Abcg2	26357	Yes	−1.32	Rnf151	67504	No	−1.13
Pla2g10	26565	Yes	−2.21	Lyzl4	69032	No	−2.26
Oaz3	53814	Yes	−5.04	Lyzl6	69444	No	−5.09
Abhd2	54608	Yes	−1.51	Tekt5	70426	No	−3.43
Ube2j1	56228	Yes	−1.7	Tssk5	73542	No	−2.26
Tbc1d20	67231	Yes	−1.46	1110017D15Rik	73721	No	−2.15
Herc4	67345	Yes	−2.68	Paqr5	74090	No	−3.12
Spaca1	67652	Yes	−4.29	Tbc1d21	74286	No	−4.62
Galntl5	67909	Yes	−6.03	Spem1	74288	No	−5.2
Syce2	71846	Yes	−1.68	Hyal5	74468	No	−4.84
Calr3	73316	Yes	−3	Spata19	75469	No	−5.1
Izumo1	73456	Yes	−1.75	Spata9	75571	No	−2.76
Spata18	73472	Yes	−2.6	Spaca3	75622	No	−1.45
Iqcf1	74267	Yes	−4.78	Sun5	76407	No	−5.08
Osbp2	74309	Yes	−2.15	Catsper3	76856	No	−5.44
Wbp2nl	74716	Yes	−3.39	4930451I11Rik	78118	No	−1.66
Ropn1	76378	Yes	−5.73	Creb3l4	78284	Yes **	−3.2
Rhbdd1	76867	Yes	−1.92	Rdh10	98711	No	−1.75
Hook1	77963	Yes	−1.43	Tubg1	103733	No	−1.14
Herpud2	80517	Yes	−1.85	Fam170b	105511	No	−2.28
Tssk6	83984	Yes	−3.53	Arrb1	109689	No	−1.05
Spag4	245865	Yes	−1.85	Plcz1	114875	No	−4.77
Chd5	269610	Yes	−2.63	Gm4787	214321	No	−4.36
Ccin	442829	Yes	−4.32	Tppp2	219038	No	−3.66
Adam4	11498	No	−3	Catsper1	225865	No	−4.38
Akap4	11643	No	−5.49	Adam6b	238405	No	−4.07
Capza3	12344	No	−5.62	Adam6a	238406	No	−3.74
Cast	12380	No	−1.75	Lrrc52	240899	No	−2.97
Cd46	17221	No	−2.31	Adcy10	271639	No	−1.6
Smcp	17235	No	−3.16	Acsbg2	328845	No	−4.72
Odf1	18285	No	−2.64	Catsper4	329954	No	−2.07
Pappa	18491	No	−1.99	Ccdc33	382077	No	−2.66
Pgk2	18663	No	−1.64	Plb1	665270	No	−1.19
Prm2	19119	No	−4.22	Prrs37	None	No	−5.55

### 4.2. Confirmed CREM Target Genes

Although the number is limited, there are studies that provide more detailed insights into the genes directly regulated by CREM, validating the data obtained through next-generation sequencing (NGS) and uncovering previously unknown gene targets. For instance, in testicular tissue, both the mRNA and protein of ATCE1/TISP40α (CREB3l4 isoform 2 or TISP40) are exclusively expressed in spermatids [57]. In vitro and in vivo analyses have demonstrated that CREMτ and TISP40 can efficiently form a heterodimer that binds to a CRE site downstream of the *Tisp40* promoter, leading to transactivation facilitated by the recruitment of the histone chaperone HIRA [58]. Thus, TISP40 is directly regulated by CREMτ and possibly participates in a positive autoregulatory loop enhanced by TISP40-CREMτ heterodimer assembly. However, the involvement of the CREMτ-TISP40 heterodimer in the differential transactivation of other genes remains unclear. Additionally, CREMτ has been shown to bind to a CRE site in the human cation channel sperm-associated 1 *CATSPER1* promoter [59] in vitro, as well as CREMτ binding to the murine *Catsper2* promoter both in vitro and in vivo [60], leading to their direct transactivation. This is consistent with the observed reduction in their mRNA levels in *Crem*-KO mice [54], indicating that *Catsper2* and likely *CATSPER1* are direct targets of CREMτ. Likewise, it has been shown that phospholipid hydroperoxide/sperm nucleus glutathione peroxidase (snPHGPx) mRNA expression is directly controlled by CREMτ [61].

Although there are numerous candidates, information regarding the identification and functional role of CREM in the direct regulation of several germline-specific gene expressions remains limited. By integrating NGS data with gain or loss of function experiments and employing proof-of-concept approaches, we are able to confidently identify these targets. Given the significance of CREM in spermatogenesis and related diseases, it is crucial to fully confirm its target genes, as this analysis will provide valuable insights into gene regulation and aid in the development of effective therapeutic strategies.

## 5. Male Fertility and CREM

Two research groups in 1996, one led by Sassone-Corsi and the other by Günther Schütz, concurrently described the first mouse model lacking *Crem*; both studies provided compelling evidence for the impact of a CREM deficiency on spermatogenesis, including the loss of the expression of vital genes for spermatogenesis, arrest at the initiation of spermiogenesis and the absence of sperm production [52,53]. The transcription factor CREM plays a central role in the formation of male haploid cells, as it is responsible for activating genes Involved in meiotic machinery and spermatogenic cell morphogenesis. In a mouse obesity model, a decrease in *Crem*, adaptor protein 1 (*Sh2b1*), desert hedgehog (*Dhh*), insulin-like growth factor 1 (*Igf1*) and leptin receptor (*Lepr*) transcript levels was observed, resulting in impaired fertility, such as a reduced sperm motility and decreased mating rates of obese males with female mice [62]. Consequently, it is plausible that CREM homologs may play a similar fundamental role in fertility in other species.

Male fertility also relies on the presence of several CREM cofactors, including ACT, KIF17b and SPAG48, which are crucial for the successful development of spermatogenic cells [12,46,47]. Studies in infertile populations have revealed the absence or low expression of CREM or its cofactors, contributing to infertility. In the case of the CREM cofactor ACT, studies in infertile patients with azoospermia or oligozoospermia have identified specific single nucleotide polymorphisms (SNPs) resulting in amino acid changes in the ACT coding region, that, combined to the haplotype 204G-211V-243R-12065G, reduce the interaction between ACT and CREM by 45%, as observed in an in vitro double hybrid test [63]. Similarly, various SNPs found in CREM and its cofactors have been linked to non-obstructive azoospermia (NOA) in patients. SNP rs4934540 in an intron region with a TT or CT genotype confers susceptibility to NOA, while a CT or CC genotype in SNP rs4934540 and an AG genotype of rs2295415 (at the 3′ untranslated end of CREM) reduce the NOA risk. The combination of four CREM SNPs in different haplotypes provides either protection (CGTG) or a high risk (TATG) for spermatogenic failure, as confirmed with expression assays, which showed a low CREM expression [64].

Patients with Klinefelter syndrome commonly experience hypogonadism, low testosterone levels and fertility problems. Testicular biopsies from these patients with mature sperm have demonstrated a lower expression of genes such as CREM and CSF-1, as well as an absence or reduced expression of protamine compared to azoospermic patients without Klinefelter syndrome but in whom complete spermatogenesis was observed histologically [65]. Most Klinefelter patients with SCO show no CREM and protamine expression with CSF-1 expression; however, CREM and protamine levels are still detectable in some patients [65]. Furthermore, alterations in the expression of activating isoforms of CREMτ, namely CREMτ1 and CREMτ2 from the P3 and P4 promoters, respectively, have been observed in patients with spermatogenesis arrest and testicular tumors [25].

Likewise, the involvement of CREM and ACT in the occupation of CRE sites in the promoters of the target genes *TNP1* and *2*, as well as *PRM 1* and *2*, in patients with the arrest of round spermatid maturation (SMA) compared to obstructive azoospermia (OA) as a positive control was evaluated. Both *CREM* and *ACT* are down-regulated in the group of SMA patients, and low levels of expression in the target genes *TNP 1* and *2* and *PRM 1* and *2* were also found. A low occupancy of CREM and ACT was also observed in the promoter regions of the *TNP1* and *2* and *PRM1* and *2* genes in the SMA group, but it could be a result of the low *CREM* and *ACT* expression [31]. This confirms CREM’s role as a transcription factor in the development of human spermatogenesis and fertility.

Moreover, the *CREM* promoter contains two CpG islands with a total of 73 CpG sites, making it susceptible to methylation. Differentially methylated sites in *CREM* have been associated with an impaired sperm DNA integrity in infertile patients [66]. However, bisulfite sequencing studies did not find significant differences in *CREM* promoter methylation in patients with oligozoospermia or abnormal protamination levels, except for two infertile patients who exhibited a distinct pattern of high methylation [40]. In contrast, a separate pyrosequencing study revealed high levels of DNA methylation in the *CREM* promoter in patients with oligozoospermia and abnormal protamination, with a negative correlation to the sperm count, morphology and motility [40]. Infection with *Toxoplasma gondii* has been shown to decrease the reproductive capacity of mice, leading to a noticeable decline in sperm production. An analysis of global methylation in the testicular tissue of infected animals revealed slight differences in methylation levels at specific sites in the *Crem* promoter [67]. These methylation-prone sites in the *CREM* promoter can modulate its expression in response to environmental or genotoxic factors, emphasizing their significance as critical determinants of sperm fertility.

## 6. Impact of Xenobiotics in CREMτ Regulation

Previously, the transcriptional regulation of *Cremτ* was discussed. However, diverse studies suggest that CREMτ is also subject to regulation by xenobiotics. Even though these foreign molecules seem to exert both positive and negative effects on fertility [41,68], extensive research needs to be performed to corroborate if the effect shown in CREMτ levels is due to a lack of a cell population expressing CREM or the direct regulation of the CREM transcript or protein.

### 6.1. CREM Disruption by ROS Production Xenobiotics

In recent years, there has been increasing evidence linking environmental pollutants to semen quality and spermatogenesis impairment, leading to a rise in male infertility rates [69,70]. Several studies have suggested that this effect is due to a down-regulation of *CREM* pollutants such as solvents, fluoride, pesticides and silica nanoparticles (SiNPs) that seem to be negative regulators of *CREM* expression [42,71,72,73].

Common pollutants like 1,2-dichloroethane (1,2-DCE), a widely used solvent, and fluoride, a prevalent contaminant in industrialized countries’ drinking water, seem to reduce *CREM* expression and its coactivator *ACT* in a dose-dependent manner. Interestingly, both pollutants are also associated with a decreased semen quality, including increased sperm malformation and a reduced concentration, viability and motility [71,72]. Additionally, 1,2-DCE exposures have been linked to the vacuolar degeneration of germ cells, sloughing of spermatogenic cells and increased apoptosis in the testes, resembling the phenotype observed in *CREM*-knockout mice [52,53].

The pesticide carbendazim (CBZ), another commonly used chemical, has been found to down-regulate *CREM* expression and decrease sperm concentration and motility. Notably, CBZ disrupts epigenetic markers such as H3K27, 5mC and 5hmC, suggesting that certain pollutants may alter the epigenetic regulation of genes involved in spermatogenesis [73]. Similarly, silica nanoparticles have been shown to reduce sperm quantity and quality by impairing the epigenetic regulation of spermatogenesis and inducing the hypermethylation of the *CREM* promoter, resulting in CREM down-regulation [41].

It is worth noting that not only pollutants but also certain medical treatments can negatively impact *CREM* expression. Cistanches herba (CH), a tonic commonly used in Eastern societies, has been found to down-regulate *CREM* expression in a dose-dependent manner while reducing the testosterone levels, sperm count and sperm motility, mirroring the phenotype observed in pollutant-induced *CREM* down-regulation [74]. Furthermore, radiation therapy, a known treatment with detrimental effects on the male reproductive system, including permanent infertility, has been shown to decrease *CREM* expression, decrease testis weight and induce atrophic seminiferous tubules [75].

The down-regulation of *CREM* by various compounds, whether pollutants or medical treatments, can impair spermatogenesis and lead to male infertility. These xenobiotics likely exert their effects through the production of high levels of reactive oxygen species (ROS), which are associated with testicular damage and male infertility (reviewed in [76]) (Figure 6A). Nevertheless, further research needs to be performed to unravel if the deregulation of CREM by these xenobiotics is because of a down-regulation of *Crem per se* or due to a lowering of cell populations expressing *Crem.*

### 6.2. CREM Up-Regulation by Antioxidants

The up-regulation of *CREMτ* has been associated with improved semen quality parameters, including sperm concentration and motility [25,62,63]. Traditional Eastern herbal remedies are promising since they have shown a positive impact on *CREMτ* expression. For example, Rubi Fructus (RF), derived from the dried fruit of *Rubus coreanus*; Yukmijihwang-tang (YJT), a multiherbal formula used to address male reproductive issues; and MYOMI, a Korean herbal medicine, have traditionally been used to enhance male fertility [77,78,79]. Studies administering these formulations to mice have demonstrated an increased sperm concentration and motility, accompanied by an enhancement in *Cremτ* mRNA and protein levels [79,80]. RF constituents are known for their antioxidant properties [75,80,81,82], while YJT and MYOMI, when tested in combination with cyclophosphamide, a commonly used chemotherapy drug, exhibited a reduced lipid peroxidation, indicating an antioxidant effect [83,84].

Antioxidant supplementation has been found to have a protective effect on male fertility [85]. For instance, lutein administration following testicular torsion reduced morphological damage to seminiferous tubules and alleviated testicular oxidative stress. Furthermore, *Cremτ* expression was restored in lutein-treated mice subjected to testicular torsion [86]. Another essential antioxidant, folic acid, a B-complex vitamin, enhanced semen quality in older roosters. Aging is known to be a factor contributing to male infertility, and folic acid supplementation increased semen volume, sperm concentration and sperm motility in older roosters. Additionally, it led to enhanced mRNA levels of important genes involved in spermatogenesis, including *Cremτ* [87]. Moreover, antioxidant supplementation in patients with idiopathic infertility resulted in the activation of proteins related to the CREM signaling pathway, such as protein kinase cAMP-dependent regulatory subunits (*PRKAR1A*, *PRKAR2A* and *PRKACA*) and lactate dehydrogenase C (*LDHC*) [88].

Taken together, these studies suggest an improvement in semen parameters and *Crem* levels due to the ability of antioxidants to quench ROS produced by different types of stress (Figure 6B)

## 7. Other CREM Implications in Health and Disease

As previously mentioned, *Cremτ* serves as a crucial regulator of spermatogenesis, but it also plays a role in various other molecular mechanisms. Several *Crem* isoforms have been identified, each with distinct functions beyond spermatogenesis. Repressive isoforms of Crem are involved in the regulation of genes associated with brain function, β cells and immune responses.

One such isoform, ICER, participates in numerous neurological processes, including long-term memory, neuronal plasticity, apoptosis and epileptogenesis [89,90,91]. Notably, ICER interacts with brain-derived neurotrophic factor (BDNF) and the signal transducer activator of transcription (STAT3) to bind to pCREB in the Gaba α1 promoter, thereby repressing its transcription in cortical neurons [92,93]. Additionally, ICER modulates adipokine production by inhibiting *Creb* and suppressing negative effectors of adiponectin, facilitated glucose transporter 4 (GLUT4) and activating transcription factor 3 (ATF3) in adipocytes [94,95]. Another metabolic pathway regulated by ICER is insulin production and secretion. ICER binds to the promoters of genes involved in the insulin pathway, and an increased ICER expression induced by oxidized LDL inhibits insulin production and secretion [96,97]. Furthermore, ICER plays a role in mediating the circadian rhythm in the liver by repressing the period circadian regulator (*Per1*) gene promoter [98]. It is also implicated in vascular smooth muscle cell apoptosis and proliferation [99].

Another repressive isoform, CREMα, is primarily involved in regulating genes related to the immune system. CREMα acts as a negative regulator of the *Cd68* gene promoter and the interleukin *Il-2* gene. Surprisingly, it increases the expression of *Il17a* through epigenetic remodeling [100,101]. CREMα is implicated in immune disorders such as Systemic Lupus Erythematosus (SLE), where its expression is enhanced by transcription factor SP1 and histone lysine methyltransferase SET1 binding to the *Crem* P1 promoter. This results in increased H3K4me3, decreased DNA methyltransferase 3a (DNMT3a) and the subsequent *CREM* methylation promoter, leading to the over-expression of CREMα [102,103]. CREMα also binds to the *Il17f* promoter and represses its expression [104], contributing to accelerated inflammation and autoimmunity [104,105].

In summary, CREM isoforms play diverse roles in various signaling pathways associated with health and disease. However, further extensive research is necessary to fully understand the involvement of other CREM isoforms in different signaling pathways.

## 8. Conclusions

The CREM gene possesses all the necessary domains to function as an activating transcription factor, specifically being involved in regulating the expression of genes crucial for spermatogenesis. However, the complexity of CREM expression arises from the presence of multiple promoters and alternative splicing, resulting in a diverse array of repressor and activator isoforms, many of which remain poorly characterized and warrant further investigation. Recent studies have shed light on the regulation of CREM at various levels, including transcriptional, post-transcriptional and post-translational regulation. Furthermore, CREM has been established as a reliable biomarker for male fertility in several toxicological studies.

Among the isoforms, the testis-specific activator isoform CREMτ exhibits distinct characteristics compared to somatic CREM isoforms. CREMτ does not rely on phosphorylation for activation and requires additional cofactor proteins for the precise temporal regulation of gene expression in post-meiotic stages. It has been shown to regulate numerous genes involved in later stages of spermatogenesis, with evidence of its binding to specific gene promoters. However, further experimental validation with an immunohistological analysis is needed to confirm the potential target genes regulated by CREMτ in testicular samples. In this regard, we summarized the potential targets of CREMτ and identified the molecular pathways they may be involved in. Additionally, we highlighted the impact of xenobiotics on CREM regulation, with a particular focus on its role in male fertility.

Nevertheless, our understanding of the molecular mechanisms governing CREM gene regulation, as well as the full scope of its isoform repertoire as important regulators of functions beyond reproduction, is still incomplete. Further research is necessary to unravel the intricate molecular mechanisms controlling CREM and its isoforms in order to fully comprehend their broader regulatory roles.

## 9. Methods and Searching Strategy

A bibliographic search was carried out using the PubMed database in February 2023 with the inclusion of documents from 2006 to the above date. Some articles before 2006 were used for introduction and historical background purposes. The terms used for the search were (CREM OR “cAMP-response element modulator”), sperm*, expression, regul*, promoter, testi*, fertil*, infertil*, toxic* and “animal model” with different combinations, and the resulting documents retrieved were manually selected, selecting only those with a direct connection to the topic.

The quaternary structure prediction of the CREMτ (NP_001104329.1) homodimer was performed using ColabFold v1.5.2: AlphaFold2 [106] with no template model, num_relax = 0, the mmseqs2_uniref_env msa mode, the unpaired_paired mode, the alphafold2_multimer_v3 model type and the remaining parameters as a default. The PDB structure was visualized and edited using ChimeraX-v1.5 for domain assignment [107]. Files generated with all predicted results can be downloaded as Appendix A.

The *Crem* expression data were obtained from the reprogenomics viewer portal (rgv.genouest.org (accessed on 19 may 2023), a repository of a reproductive cell data set [108,109], from a study by Green et al., 2018 [28], a transcriptome study that analyzed the sequences of 34,633 cells using the single-cell sequencing (scRNA-Seq) of C57BL/6J male mice during normal spermatogenesis in adults (7–9 weeks). The file called “data_genelevel_Adult_germ_cells_Fig2A” was taken because it considers the entire spectrum of germ cells with N = 20,646 cells, without somatic cells. The *Crem* (12,916) gene was selected to analyze its expression, and visualization was performed in a violin plot with the class selection as the cell type to generate Figure 3. Data are plotted as an expression intensity of fragments per kilobase of the transcript per million fragments (FPKM), according to results of the StringTie program. 

The gene-ontology–biological-process analysis was performed using ShinyGO 0.77 [110] with FDR cut-off = 0.05, selecting mouse species, removing redundancy and using the pathway database for the gene count reference. The gene list submitted to the GO analysis was extracted, applying a cut-off in log2FC of 1 regarding differentially expressed genes in mice CREM-KO testis samples from GSE29593 [55].

Figure 2, Figure 4 and Figure 6 were made using Inkscape 1.2. All figures presented here were made for this review article using data mentioned above.

## Figures and Tables

**Figure 1 ijms-24-12558-f001:**
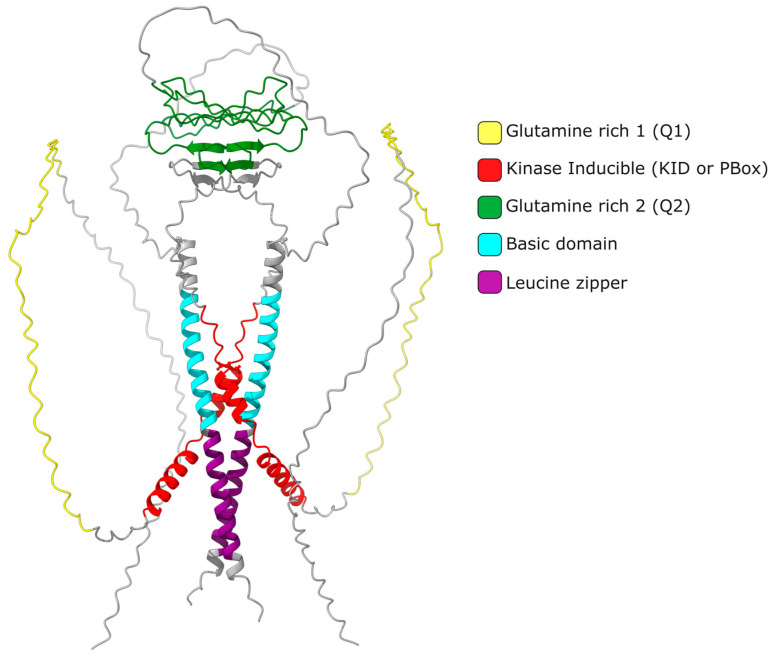
Structure of Cremτ homodimer: The predicted quaternary structure of the cAMP-response element modulator CREMτ homodimer is displayed. Domains are highlighted in different colors. The Q1 and Q2 domains have the ability to interact with proteins of the basal transcription machinery as well as other proteins. Within the kinase-inducible Kid domain, several residues (e.g., S117) can undergo phosphorylation by proteins with kinase activity, such as testis-specific serine kinase (TSSK4) and cAMP-dependent protein kinase (PKA), thereby enhancing their activity. The basic domain is responsible for DNA binding and allows for the recognition of CRE sites. The leucine zipper domain facilitates both hetero- and homodimerization. The basic domain and leucine zipper are commonly abbreviated as the bZIP domain. Regions in grey color are not considered as belong to any domain. Structure prediction was performed using AlphaFhold2 with a CREMτ (NP_001104329.1) mouse AA sequence.

**Figure 2 ijms-24-12558-f002:**
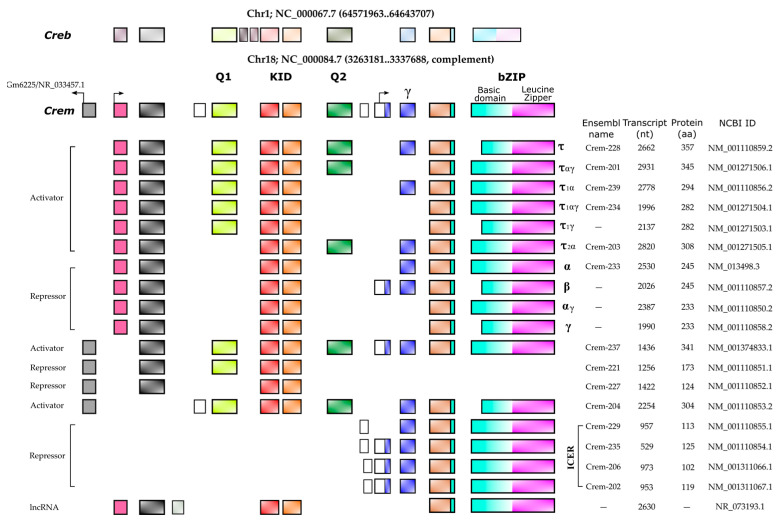
Mouse *Crem* isoforms. The structure of 18 confirmed isoforms at transcriptional and protein levels is shown. The mouse *Crem* gene is located on chromosome 18 in the complementary strand where two lncRNAs (NR_073193.1 and Gm6225) are located in the sense and antisense direction, respectively. The *Creb* gene is shown above for comparison. The CREM domains in the gene structure are highlighted in different colors at the top and the composition of each isoform is shown. For identification, the Ensembl name of each isoform, the length of the gene in nucleotides (nt), the protein length in amino acids (aa) and the corresponding NCBI ID are listed.

**Figure 3 ijms-24-12558-f003:**
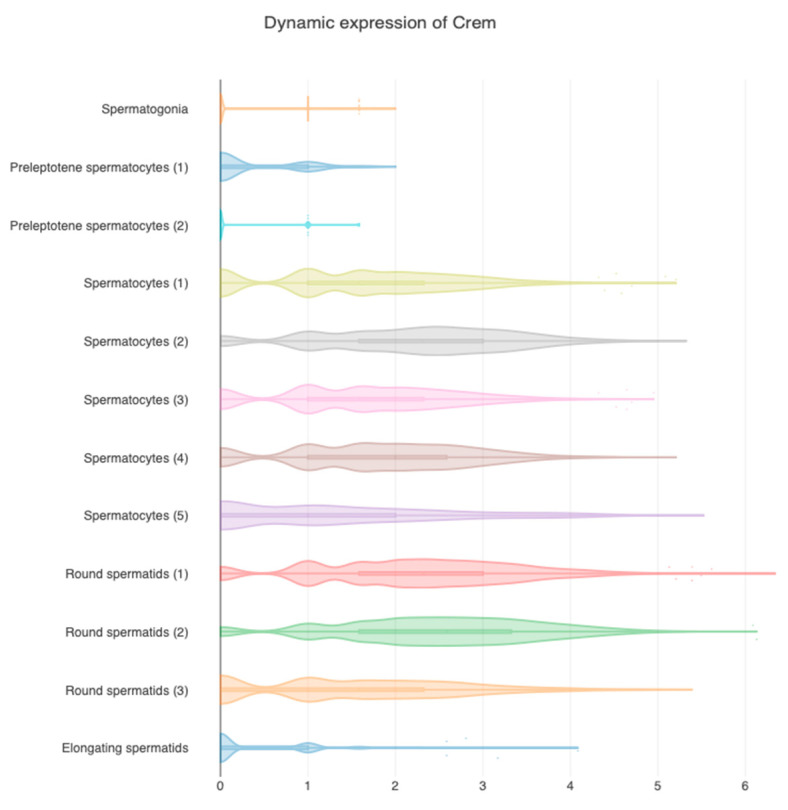
*Crem* expression at different stages of spermatogenesis. The scRNAseq data set from germ cells of male C57BL/6J mice, in which the stages of sperm formation were assigned using cell-type-specific markers [28]. The *Crem* expression query indicates different expression levels for each stage of spermatogenesis. The highest expression is observed in round spermatids, and the lowest in preleptotene spermatocytes (1 and 2). Numbers on the *x*-axis indicate expression intensity as fragments per kilobase of the transcript per million fragments (FPKM). The figure was made using the reprogenomics viewer portal (rgv.genouest.org) with data from [28].

**Figure 4 ijms-24-12558-f004:**
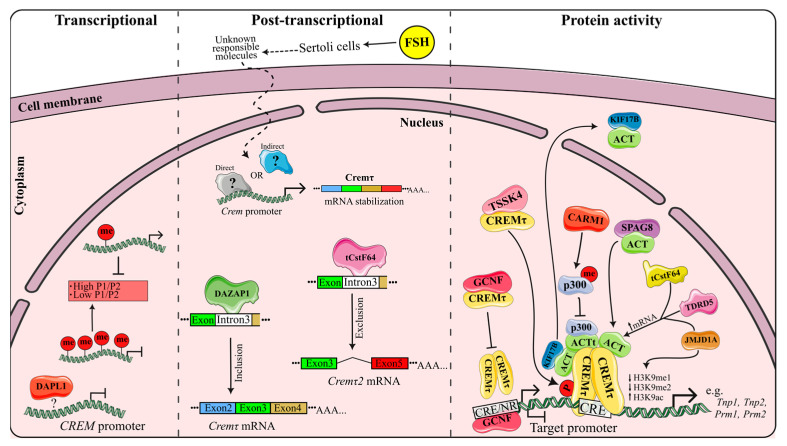
Mechanisms of Crem regulation. *Transcriptional regulation of Crem* (***left panel***): Hypermethylation of the *Crem* promoter leads to a reduction in CREM expression, resulting in the decreased expression of protamine genes and, consequently, an abnormal protamination is produced. Death-Associated Protein-like 1 (DAPL1) is able to reduce the expression of *Crem* transcripts through an unknown mechanism. *Post-transcriptional regulation of Crem* (***middle panel***): Sertoli cells recognize Follicle-stimulating hormone (FSH), and through an unknown mechanism, induce the stabilization of *Cremτ* mRNA in germ cells. Deleted in azoospermia-associated protein 1 (DAZAP1) and stimulatory cleavage factor (τCSTF-64) can bind to newly synthesized mRNA in *Crem* intron 3, mediating the inclusion and exclusion of *Crem* exon 4 (Q1 domain), respectively. *Regulation of CREM protein activity* (***right panel***): H3K9me1/H3K9me2 lysine demethylase JMJD1a, τCSTF-64 and Tudor domain protein 5 (TDRD5) indirectly modulate the activity of CREM protein, either enhancing or maintaining the expression of its coactivator ACT. JMJD1a also maintains low methylation levels on promoters of the *Crem* target gene, facilitating the correct positioning of CREM. TSSK4 interacts with CREM and phosphorylates its S117, increasing CREM activity. Protein arginine methyltransferase (CARM1) methylates p300, preventing it from interacting with ACT and reducing CREMτ transactivation. Sperm-associated antigen 8 (SPAG8) interacts with ACT, aiding in the binding of ACT with CREMτ, but without forming a SPAG8-ACT-CREMτ complex, thereby increasing the transcriptional activity of CREMτ. GCNF interacts and competes with CREMτ for binding to CRE/NR sites, suppressing its activity. Lastly, kinesin protein (KIF17b) “kidnaps” ACT, taking it to the cytoplasm and preventing ACT-CREMτ interaction.

**Figure 5 ijms-24-12558-f005:**
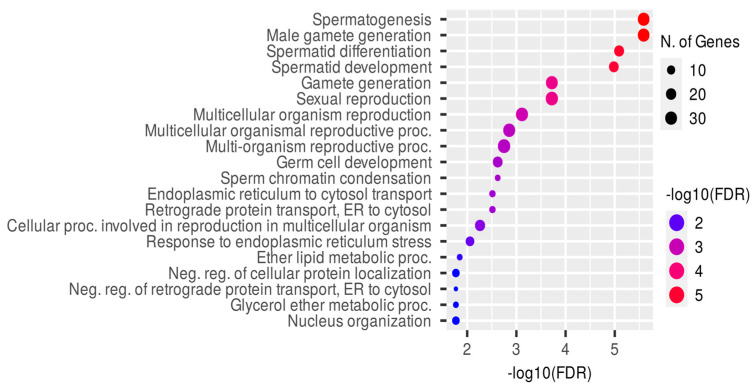
Gene ontology (GO) of down-regulated genes bound by CREM. The top 20 gene ontology results of the group of down-regulated genes in the *Crem*-KO testis known to be bound by CREM show an enrichment of the processes related to spermatogenesis and sexual reproduction, but other processes are mainly related to protein transport. The gene ontology analysis was performed using ShinyGO 0.77.

**Figure 6 ijms-24-12558-f006:**
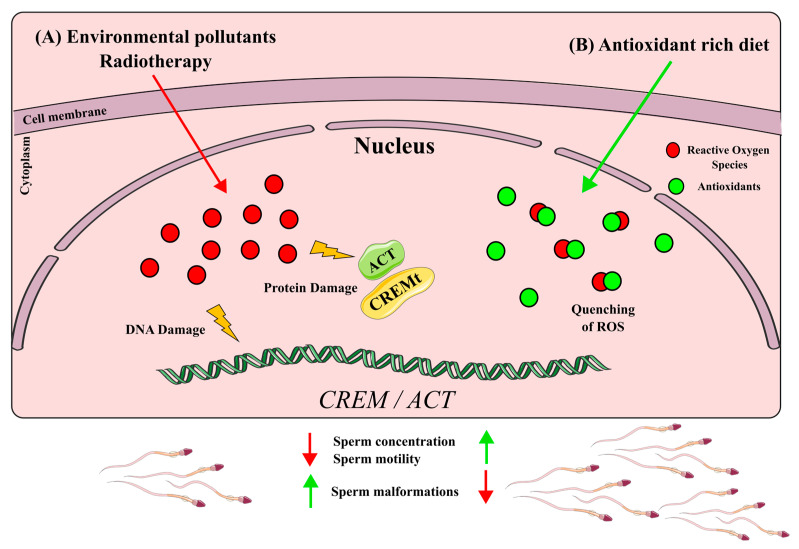
Effect of xenobiotics on *Crem* regulation. (**A**) Xenobiotics that produce ROS down-regulate *CREM/ACT* expression and damage DNA and protein. This affects sperm quality and reduces sperm concentration and motility. (**B**) Antioxidants quench ROS, enhancing *Crem* expression. This is associated with better semen quality parameters such as sperm concentration and motility.

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
