# Peer review of "Novel Aspects of cAMP-Response Element Modulator (CREM) Role in Spermatogenesis and Male Fertility"

_ijms, 2023, doi:10.3390/ijms241612558_

Round 1

Reviewer 1 Report

Comments on the review article entitled

Novel aspects of cAMP-response element modulator (CREM) role in spermatogenesis and male fertility.

The authors review the newer literature on the role of the transcription factor CREM in (mammalian) spermatogenesis and male fertility. Generally, the main findings on the regulation and function of CREM during mammalian spermatogenesis have been made 15 to 30 years ago. In most papers thereafter, CREM has been used only as a marker for testicular meiotic (RNA level) or postmeiotic (protein level) germ cells. Therefore, the increase in knowledge from after ~ 2010 is rather small. In light of this, the choice of 2006 or newer (the hot phase of CREM research started in 1992) as an inclusion criterion is questionable, as papers really essential to the topic of the review and frequently cited were surprisingly not discussed. Nevertheless, a review on the regulation and function of CREM in mammalian spermatogenesis is currently useful because no work has systematically reviewed the CREM field in recent years.

However, there are some points that I believe should or need to be revised before publication.

1.       The review contains numerous statements that are made without reference. This is especially true, but not limited to, the introductory passages. Just as an example, but not limited to: Lines 555-564.

2.       This manuscript is a mixture of review and own evaluations / data. It should be clearly idicated which tables and graphs were created by the authors for this manuscript and what, if any, was taken from other papers.

3.       I suggest to remove chapter 7.7 from the manuscript, because it goes beyond the title of the review and does not fit into the train of thoughts in my opinion.

4.       Unfortunately, in publications on spermatogenesis research, one finds again and again a fallacy. Testicular tissue samples are analyzed by PCR or other RNA quantification methods without histological analysis having taken place. Often the cellular composition of the examined testes (infertile males or hormonally or by toxins / environmental factors reduced spermatogenesis in experimental animals) is significantly altered compared to the healthy / untreated control. This often leads to the erroneous conclusion that experimental treatment regulates CREM expression. However, often only the cell population expressing CREM is no longer present, so that less CREM transcript is detectable in the entire testis because of this. A well-known example of presumably this misinterpretation is the work of Foulkes et al, which is also important for this review (Nature. 1993, Pituitary hormone FSH directs the CREM functional switch during spermatogenesis). It stands to reason, since there was no histological analysis of the testis in that paper, that simply the germ cell stages expressing CREMτ were largely or completely released from the germinal epithelium after one or two weeks without any FSH (hypophysectomy) and therefore the activator isoform was no longer detectable. Therefore, it is always of utmost importance to determine the cellular composition in testicular tissue samples in which CREM (or also other factors, e.g. Ref 34) are to be quantified. Only when CREM shows differences in expression levels in individual cells under treatment is it justified to speak of a treatment regulating CREM (and not the CREM expressing cell population). All papers mentioned in the review should be checked for this criterion and discussed accordingly (e.g. Ref 30, 62, 69, ...).

5.       In this respect, I also believe that the interpretation that FSH controls the CREM switch is a misinterpretation. That FSH induces ICER in Sertoli cells, on the other hand, is credible in my view.

6.       I am surprised that the authors cite publications like reference 69. The histological data of this publication including CREM immunohistochemistry are so weak that they should not be cited in my estimation. I at least am not able to recognize the CREM IHC signals shown there, and quantitative evaluation of this staining seems impossible to me.

Minor Things:

Line 36: Leydig cells also express the AR.

Line 42: Genes are transcribed and transcripts are translated.

Line 72/73: ACT stands for Activator of Crem in testis

Line 90/91: see above

Figure 2: explanation of the diagram (scale / units of expression levels)

Line 141: name of CREM gene

Lines 223/224: isn’t is surprising that the mediator of FSH action (in Sertoli cells) on CREM expression (in germ cells) has not been identified so far? For me, it is an indicator that the whole FSH / CREM switch-story is a myth.

Line 437: Wouldn’t it be more informative to do ChIP-Seq analyses in wildtype mice (than in crem-mutant mice)?

Table 1: are these “deregulated genes” really regulated by CREM or do they show differential abundance due to different cell compositions of the testes analyzed? (Compare major item 4)

Line 601/602: high sperm concentration suggests high spermatid concentration in testis samples > high CREM activator abundance > testicular cell composition matters.

I am not a native speaker myself. But my impression is that language editing would be useful.

Author Response

Reviewer 1.

Comments on the review article entitled

Novel aspects of cAMP-response element modulator (CREM) role in spermatogenesis and male fertility.

The authors review the newer literature on the role of the transcription factor CREM in (mammalian) spermatogenesis and male fertility. Generally, the main findings on the regulation and function of CREM during mammalian spermatogenesis have been made 15 to 30 years ago. In most papers thereafter, CREM has been used only as a marker for testicular meiotic (RNA level) or postmeiotic (protein level) germ cells. Therefore, the increase in knowledge from after ~ 2010 is rather small. In light of this, the choice of 2006 or newer (the hot phase of CREM research started in 1992) as an inclusion criterion is questionable, as papers really essential to the topic of the review and frequently cited were surprisingly not discussed. Nevertheless, a review on the regulation and function of CREM in mammalian spermatogenesis is currently useful because no work has systematically reviewed the CREM field in recent years.

In advance, we appreciate your detailed comments and suggestions. It certainly will be helpful for the quality enrichment of this review.

The aim of this article is to examine the new aspects of the role of CREM in spermatogenesis and male fertility. As you mentioned, most of the knowledge about the involvement of CREM in spermatogenesis was already gathered from 1992 to around 2004. At that time, the most important basic research on CREM was carried out, with several review articles being published that better describe the essential aspects, as the authors were the main contributors to CREM research. In fact, the last review in the CREM field was carried out in 2006, summarizing the knowledge gained up to that point. The reader could easily find these articles in the bibliography section as prompted in the main text. Based on this, we have scrupulously chosen to include work from 2006 to date that contains new information reported in the CREM area, with particular attention to its role during spermatogenesis, in order to better understand its importance and progress, identify unanswered questions or delineate areas of application, and suggest ways of approach. We included a paragraph in the introduction to inform the reader about this situation in lines 86-90.

However, there are some points that I believe should or need to be revised before publication.

  1. The review contains numerous statements that are made without reference. This is especially true, but not limited to, the introductory passages. Just as an example, but not limited to: Lines 555-564.

Answer to concern 1: The manuscript was revised for these statements and the references have been added and highlighted in lines 220, 573-578 and 617.

  1. This manuscript is a mixture of review and own evaluations / data. It should be clearly indicated which tables and graphs were created by the authors for this manuscript and what, if any, was taken from other papers.

Answer to concern 2: All figures were originally made for this review but figures 2, 5 and table 1 were created from previously published (publicly available) data whose references are properly cited in the text lines 67,178 and methods in lines 741-742. Nevertheless, not all of the data in the original research article was used to generate the figures, only those that met the established criteria. See Section 9 (Methods and Search Strategy) for detailed inclusion criteria, data source and software used for analysis. You can also find the complete results as supplementary files.

.

  1. I suggest to remove chapter 7.7 from the manuscript, because it goes beyond the title of the review and does not fit into the train of thoughts in my opinion.

Answer to concern 3: In fact, it is unrelated to the title of the review, but the goal of this chapter is to briefly illustrate that CREM is involved in many health and disease signaling pathways, not just spermatogenesis. In addition, we wish to describe the effects of other previously described CREM isoforms and emphasize that CREM is not the only isoform of the gene involved in gene regulation. This also highlights that CREM research has recently expanded to other tissues and that there is an association with diseases other than spermatogenesis.

  1. Unfortunately, in publications on spermatogenesis research, one finds again and again a fallacy. Testicular tissue samples are analyzed by PCR or other RNA quantification methods without histological analysis having taken place. Often the cellular composition of the examined testes (infertile males or hormonally or by toxins / environmental factors reduced spermatogenesis in experimental animals) is significantly altered compared to the healthy / untreated control. This often leads to the erroneous conclusion that experimental treatment regulates CREM expression. However, often only the cell population expressing CREM is no longer present, so that less CREM transcript is detectable in the entire testis because of this. A well-known example of presumably this misinterpretation is the work of Foulkes et al, which is also important for this review (Nature. 1993, Pituitary hormone FSH directs the CREM functional switch during spermatogenesis). It stands to reason, since there was no histological analysis of the testis in that paper, that simply the germ cell stages expressing CREMτ were largely or completely released from the germinal epithelium after one or two weeks without any FSH (hypophysectomy) and therefore the activator isoform was no longer detectable. Therefore, it is always of utmost importance to determine the cellular composition in testicular tissue samples in which CREM (or also other factors, e.g. Ref 34) are to be quantified. Only when CREM shows differences in expression levels in individual cells under treatment is it justified to speak of a treatment regulating CREM (and not the CREM expressing cell population). All papers mentioned in the review should be checked for this criterion and discussed accordingly (e.g. Ref 30, 62, 69, ...)

Answer to concern 4: The reviewer's point of view is correct, the histological changes due to the treatments are not taken into account and there are no results in this sense that would allow a correct interpretation of the Crem expression. The fact that the Crem-expressing cell population was reduced could be due to the down-regulation of Crem per se, as shown in the CREM K.O. mice in which post-meiotic arrest occurs, thereby reducing the cell population expressing the activator isoform of CREM (Nantel et al., 1996). However, round spermatids are still present in the testis of the Crem-deficient mouse model, a population of cells positively immunodetecting Crem in wild-type (Behr & Weinbahuer, 2001) and expression by scRNAseq (Figure 3), but postmeiotic arrest reduced the elongated spermatid population, neglecting Crem expression. In this regard, this histological analysis does not help to determine whether CREM is reduced due to the absence of the cell population-expressing CREM or whether the absence of this population is due to the down-regulation of CREM per se. About this concern, a paragraph was included in lines  237 -241.

 This, combined with the fact that human testicular biopsies are difficult to obtain, hampers research into male fertility-related genes or therapies. However, Sertolli cell-only syndrome (SCOS), arrested round spermatid maturation (SMA) and obstructive azoospermia (OA) required testicular biopsy and histological analysis to diagnose such disorders and certainly resulted in CREM expression in these diagnosed patients [30]. Now this is explained in lines 191-196.  In Klinefelter patients (KS), all TESE samples underwent histological analysis to classify them as complete spermatogenesis (CS), KS with mature sperm as a mixed of focal spermatogenesis and hyalinized tubules (MFSHT) or KS without mature sperm as SCO with contradictory CREM expression results [62] that are mentioned in lines 540-543.

To date most of these research studies examining the influence of xenobiotics on sperm parameters use CREM as a marker for cell differentiation. However, more extensive studies need to be done to elucidate if these molecules have an impact on

CREM expression or the cell population-expressing CREM. The text has been modified to discuss better the references relating to this topic and to address the concerns of the reviewer in lines 612-614.

In future studies it would be advisable to observe the histological changes in the testicular tissue that support the presence of cells that normally express Crem.  This was included in lines 700-702.

  1. In this respect, I also believe that the interpretation that FSH controls the CREM switch is a misinterpretation. That FSH induces ICER in Sertoli cells, on the other hand, is credible in my view.

Answer to concern 5: Indeed, extensive research needs to be done to definitively elucidate if FSH controls the CREM switch or not, as stated in lines 231-233, 237-241. Modifications in the text have been made to support this statement in other sections in lines 103,226.

  1. I am surprised that the authors cite publications like reference 69. The histological data of this publication including CREM immunohistochemistry are so weak that they should not be cited in my estimation. I at least am not able to recognize the CREM IHC signals shown there, and quantitative evaluation of this staining seems impossible to me.

Answer to concern 6: As shown in Figures 3 and 4 of the reference, there seems to be no difference in the histological data nor in sperm count, even though they show a significant difference in this last parameter, of the group treated with 50 mg/L NaF and the control, nevertheless, is a significant decreased in Crem and Act transcripts levels (approximately 40% of reduction). This downregulation in mRNA probably affects the protein level of both genes; even this is not very well demonstrated in the immunohistochemistry. The text has been modified to add the proper questions about the data presented in this reference (lines 586-587, 612-614).

Minor Things:

Line 36: Leydig cells also express the AR.

Answer to minor concern 1: Now, in line 36, Leydig cells are also mentioned as an expressed androgen receptor.

Line 42: Genes are transcribed and transcripts are translated.

Answer to minor concern 2: The mistake was corrected in line 42, thanks for the observation.

Line 72/73: ACT stands for Activator of Crem in testis

Answer to minor concern 3: Done, the name has been modified in line 74.

Line 90/91: see above

Answer to minor concern 4: The misinterpretation was changed in line 103.

Figure 2: explanation of the diagram (scale / units of expression levels)

Answer to minor concern 5: This was an undesirable omission, thanks for pointing it.  The number in x-axis indicates expression intensity as fragments per kilobase of transcript per million fragments (FPKM) according to results of StringTie program. This is indicated now in the lines 176-177 of figure 3 and explained in methods line 733-735.

Line 141: name of CREM gene

Answer to minor concern 6: Now, only gene symbol is written in line 143, as the complete name was aforementioned in line 44.

Lines 223/224: isn’t is surprising that the mediator of FSH action (in Sertoli cells) on CREM expression (in germ cells) has not been identified so far? For me, it is an indicator that the whole FSH / CREM switch-story is a myth.

Answer to minor concern 7: Yes, it is. We think part of the issue would be easily solved nowdays if someone be interested in. However, there is not enough available evidence to fully take a stand. For now, we can only highlight data supporting and opposing both sides of the story. Lines 231-235 and 237-241.

Line 437: Wouldn’t it be more informative to do ChIP-Seq analyses in wildtype mice (than in crem-mutant mice)?

Answer to minor concern 8: There was a typing omission but yes, of course. Indeed, there is no logical reason to conduct a ChIP-seq analysis of Crem in a Crem-KO sample. Now, it is corrected in lines 458-459.

Table 1: are these “deregulated genes” really regulated by CREM or do they show differential abundance due to different cell compositions of the testes analyzed? (Compare major item 4)

Answer to minor concern 9: Well, that’s the main problem of conducting an expression analysis in KO models without taking into account the resulting phenotype, in this case, arrest on early round spermatids (complete lacking elongated spermatids). It is especially true when downregulated genes in Crem-KO testis are originally expressed in late spermatids stages (e.g. Prm1 and Oaz3) but not when expressed in earlier stages. Nevertheless, Chip-seq analysis shows that some of these post meiotic genes (e.g. Prm1 and Oaz3) are occupied by Crem in wt, suggesting a regulatory role of Crem in these post meiotic genes. In these cases, it is impossible to determine a causal relationship with these methods. These concerns are strikingly pointed in the main text (section 4.1) as well as plausible solutions or suggestions (quite good example of this rationale could be find in lines 469-477). Without mention that for several of these genes, we do not know with accuracy in which cell type its expression begins. On the whole, these concerns are the reason for naming table 1 as “Deregulated genes involved in spermatogenesis process from Crem KO testis” and not “Genes involved in spermatogenesis regulated by Crem”. The same issue applies for figure 5, and this concern is pointed too. In direct response to your question, deregulated genes are a consequence for both situations.

Line 601/602: high sperm concentration suggests high spermatid concentration in testis samples > high CREM activator abundance > testicular cell composition matters.

Answer to minor concern 10: this rational was used in a brief conclusion in lines 641-644 at the end of paragraph.

Comments on the Quality of English Language

I am not a native speaker myself. But my impression is that language editing would be useful.

A careful review was made to improve the spelling of some sentences that were highlighted

Reviewer 2 Report

Everything you always wanted to know about Crem isoforms is in this review.  This well written, comprehensive and critical evaluation of the field is a valuable contribution to understanding the multiplicity of functions including transcriptional activity and protein protein interactions involved in spermatogenesis.  From this review it is abundantly clear that the CREM gene possesses all the necessary domains to function as an activating tran-  scription factor, specifically involved in regulating the expression of genes crucial for spermatogenesis.  Also, for future studies unanswered questions are clearly recognized.

Language quality is excellent.  There are a few places where editing is required.

Author Response

Reviewer 2

Comments and Suggestions for Authors

Everything you always wanted to know about Crem isoforms is in this review.  This well written, comprehensive and critical evaluation of the field is a valuable contribution to understanding the multiplicity of functions including transcriptional activity and protein interactions involved in spermatogenesis.  From this review it is abundantly clear that the CREM gene possesses all the necessary domains to function as an activating transcription factor, specifically involved in regulating the expression of genes crucial for spermatogenesis.  Also, for future studies unanswered questions are clearly recognized.

Answer to reviewer 2

Answer to reviewer: We appreciate so much your commentary about this review, this is the result of a great effort from our students who are experimentally working with Crem regulation.  We have detected the sentences that need to be corrected for better quality, which were highlighted.

Reviewer 3 Report

  • The author systematacially reviewed the roles of CREM in spermatogenesis and male fertility, which will help us understand the research progress and deficiency of CREM. However, I have some concerns about current edition.

    1. Line 44, the full name of Crem should be written at first time. Similar situations should be checked for the full text.

    2. In the figure 6, I don’t see the symbol ‘A’ and ‘B’.

    3. In line 497-499, in the mouse obesity model, 5 genes were downregulated, including Crem, Sh2b1, Dhh, Igf1, and Lepr. Crem may be important, but other genes also contribute, so you need adjust your expression.

    4. In line 633, the serial number of the title may be wrong, please check carefully. In addition, is this part of the summary necessary? After all, these diseases have no relationship with spermatogenesis and male infertility. The title is about spermatogenesis and male fertility, though as extension, other health disorders about CREM should not introduce such much.

    5. The part of conclusion briefly presents the current state of research and describes the scientific problems to be studied. Since CREM has been involved in clinical studies, do you think CREM has potential in clinical diagnosis and treatment in the future?

    6. According to your references, there has not been much research related to CREM in the last 5 years, what do you think is constraining scientists to study this topic?

    7. In Figure1, the color used in the picture and the color labels are not same.

    8. In Figure2, which standard did it used? what’s the meaning of “1,2,3”?

    9. In Figure5, the size and format of front need change.

    10. Line 508&706, something is wrong with type setting.

    11. Line 190,what does the punctuation beforefactorsmean?

English language is good, however, some spelling and sentences need to be reconsidered.

Author Response

Comments and Suggestions for Authors

The author systematically reviewed the roles of CREM in spermatogenesis and male fertility, which will help us understand the research progress and deficiency of CREM. However, I have some concerns about current edition.

Answer to reviewer: We are grateful to reviewer 2 for the critical review and suggestions for this article.

Concerns:

  1. Line 44, the full name of Crem should be written at first time. Similar situations should be checked for the full text.

Answer to concern 1: The full names of Crem and other mentioned genes throughout the manuscript were added in lines 44,63,74,100, 165, 232,335,341,344,389,423-425,491,516,638, 660, 664, 669, 676. Now the manuscript becomes more understandable, thanks for your observation.

  1. In the figure 6, I don’t see the symbol ‘A’ and ‘B’.

Answer to minor concern 2: The symbol A and B has been added to the figure 6 as mentioned in the text.

  1. In line 497-499, in the mouse obesity model, 5 genes were downregulated, including Crem, Sh2b1, Dhh, Igf1, and Lepr. Crem may be important, but other genes also contribute, so you need adjust your expression.

Answer to minor concern 3: Reviewer is right about these other genes could affect also fertility, so the sentence has been modified to comprise this possibility. Now in lines 516-517.

  1. In line 633, the serial number of the title may be wrong, please check carefully. In addition, is this part of the summary necessary? After all, these diseases have no relationship with spermatogenesis and male infertility. The title is about spermatogenesis and male fertility, though as extension, other health disorders about CREM should not introduce such much.

Answer to minor concern 4: The serial number of the title was corrected in line 652. The aim of this topic is to illustrate to the reader that CREM participates in many pathways in health and disease, not only spermatogenesis. Moreover, our intention is to describe the implications of other CREM isoforms described earlier in the text.

  1. The part of conclusion briefly presents the current state of research and describes the scientific problems to be studied. Since CREM has been involved in clinical studies, do you think CREM has potential in clinical diagnosis and treatment in the future?

Answer to minor concern 5: CREM is widely used as a differentiation marker for post-meiotic stages of spermatogenesis and has been related with male fertility parameters when an experimental strategy is conceivable (mouse or other models), as described in the text. Nowadays there are just a couple reports of mutations or SNPs in the CREM gene that could be used as a biomarker for human male fertility or a deregulation of its expression in infertile males, but this option could not be discarded. In our opinion, this approach is the possible one when human diagnosis is required without taking biopsy samples. As for a possible treatment, yes, but so far only indirect modulation of Crem expression in human has been reported (through medical treatments or some herbal remedies). Future research needs to be conducted to unravel and confirm the importance of Crem for the clinical diagnosis and treatment of infertile men.

  1. According to your references, there has not been much research related to CREM in the last 5 years, what do you think is constraining scientists to study this topic?

Answer to minor concern 6: Some of the essential mechanisms in which CREM is implicated in in spermatogenesis have been described mainly 90’s. Since the research of gene regulation nowadays is focused on Genome-Wide Analysis and epigenetics mechanisms, CREM research has lost interest in the field due to the discovery of new fertility targets; however, basic regulation of Crem expression has not been unraveled and the role of its multiple isoforms is pending to resolve.

  1. In Figure1, the color used in the picture and the color labels are not same.

Answer to minor concern 7: Thank you for the observation, in this occasion the color labels of the Figure1 has been corrected to have the same solid color.

  1. In Figure2, which standard did it used? what’s the meaning of “1,2,3…”?

Answer to minor concern 8: This was an undesirable omission, thanks for pointing it. There is not a standard as comparison reference for Crem expression. The number in x-axis indicates expression intensity as fragments per kilobase of transcript per million fragments (FPKM) according to results of StringTie program. This is indicated now in the lines 176-178 and explained in methods line 733-735. Figure 2 changes to figure 3 in this version.

  1. In Figure5, the size and format of front need change.

Answer to minor concern 9: Thanks for the observation, now the size is reduced. Unfortunately, the software using a specific font to generate the diagram, so we cannot change it.

  1. Line 508&706, something is wrong with type setting.

Answer to minor concern 10: Type setting from 508 was corrected in current lines 526-528. There is nothing wrong with the type setting of 706. The justified format does the automatic spacing due to the large name of file “data_genelevel_Adult_germ_cells_Fig2A”, which cannot be modified, as it’s the original name’s file.

  1. Line 190,what does the punctuation before“factors”mean?

Answer to minor concern 11: It was a character added by mistake, now the paragraph in lines 191-197 was changed due a concern of revisor1.

Comments on the Quality of English Language

English language is good, however, some spelling and sentences need to be reconsidered.

A careful review was made to improve the spelling of some sentences that were highlighted

Round 2

Reviewer 1 Report

Thank you authors for your constructive and clear responses to my comments and questions. Most of my points have been reasonably addressed in my view and the manuscript modified accordingly.

Given the title of the review, I would still recommend to remove Paragraph 7 from the manuscript. However, this is only as a recommendation; the decision should be made by the authors.

Finally, I would ask the authors to check / correct the following (mostly minor) points:

Line 35: as the AR is an intracellular receptor I would recommend to replace “on” with “in”.

Line 40: check sentence.

Lines 44, 70 and throughout the MS: check gene / protein nomenclature of mammalian genes / proteins.

Line 40: display

Line 57: CREB stands for cAMP responsive element binding protein

Line 88: describe essential effects

Legend Fig.3: Ref 18 is apparently not correct. Double-check references of the revised version of the MS.

Lines 229-241: the inclusion of this paragraph (including the highly relevant reference on the FSH-R knockout mice that are still fertile, i.e. display a different phenotype than CREM mutant mice; the reference on the fertile FSH knockout males could also be included: https://pubmed.ncbi.nlm.nih.gov/9020850/  ) is appreciated. However, it is recommended to rephrase the sentence in lines 230-233.

Proofreading by a native speaker would be recommended. 

Reviewer 3 Report

The MS has been improved properly.

No comments.